# Evaluation of Artificial Intelligence-Calculated Hepatorenal Index for Diagnosing Mild and Moderate Hepatic Steatosis in Non-Alcoholic Fatty Liver Disease

**DOI:** 10.3390/medicina59030469

**Published:** 2023-02-27

**Authors:** Zita Zsombor, Aladár D. Rónaszéki, Barbara Csongrády, Róbert Stollmayer, Bettina K. Budai, Anikó Folhoffer, Ildikó Kalina, Gabriella Győri, Viktor Bérczi, Pál Maurovich-Horvat, Krisztina Hagymási, Pál Novák Kaposi

**Affiliations:** 1Medical Imaging Center, Department of Radiology, Faculty of Medicine, Semmelweis University, Korányi S. u. 2/A., 1083 Budapest, Hungary; 2Department of Internal Medicine and Oncology, Faculty of Medicine, Semmelweis University, Korányi S. u. 2/A., 1083 Budapest, Hungary; 3Department of Surgery, Transplantation and Gastroenterology, Faculty of Medicine, Semmelweis University, Üllői út 78., 1082 Budapest, Hungary

**Keywords:** ultrasound, liver, artificial intelligence, non-alcoholic fatty liver disease, hepatorenal index

## Abstract

*Background and Objectives*: This study aims to evaluate artificial intelligence-calculated hepatorenal index (AI-HRI) as a diagnostic method for hepatic steatosis. *Materials and Methods*: We prospectively enrolled 102 patients with clinically suspected non-alcoholic fatty liver disease (NAFLD). All patients had a quantitative ultrasound (QUS), including AI-HRI, ultrasound attenuation coefficient (AC,) and ultrasound backscatter-distribution coefficient (SC) measurements. The ultrasonographic fatty liver indicator (US-FLI) score was also calculated. The magnetic resonance imaging fat fraction (MRI-PDFF) was the reference to classify patients into four grades of steatosis: none < 5%, mild 5–10%, moderate 10–20%, and severe ≥ 20%. We compared AI-HRI between steatosis grades and calculated Spearman’s correlation (r_s_) between the methods. We determined the agreement between AI-HRI by two examiners using the intraclass correlation coefficient (ICC) of 68 cases. We performed a receiver operating characteristics (ROC) analysis to estimate the area under the curve (AUC) for AI-HRI. *Results*: The mean AI-HRI was 2.27 (standard deviation, ±0.96) in the patient cohort. The AI-HRI was significantly different between groups without (1.480 ± 0.607, *p* < 0.003) and with mild steatosis (2.155 ± 0.776), as well as between mild and moderate steatosis (2.777 ± 0.923, *p* < 0.018). AI-HRI showed moderate correlation with AC (r_s_ = 0.597), SC (r_s_ = 0.473), US-FLI (r_s_ = 0.5), and MRI-PDFF (r_s_ = 0.528). The agreement in AI-HRI was good between the two examiners (ICC = 0.635, 95% confidence interval (CI) = 0.411–0.774, *p* < 0.001). The AI-HRI could detect mild steatosis (AUC = 0.758, 95% CI = 0.621–0.894) with fair and moderate/severe steatosis (AUC = 0.803, 95% CI = 0.721–0.885) with good accuracy. However, the performance of AI-HRI was not significantly different (*p* < 0.578) between the two diagnostic tasks. *Conclusions*: AI-HRI is an easy-to-use, reproducible, and accurate QUS method for diagnosing mild and moderate hepatic steatosis.

## 1. Introduction

Non-alcoholic fatty liver disease (NAFLD) is the most common cause of chronic liver disease in Western countries, and it has a 25% prevalence worldwide [1]. There is a strong association with type 2 diabetes; NAFLD is a frequent indication for liver transplantation and a significant cause of cardiovascular morbidity. Fat accumulation in ≥5% of the hepatocytes detected either by histology, or magnetic resonance imaging (MRI) is a prerequisite to the NAFLD diagnosis. To facilitate early diagnosis and prevent complications from NAFLD, current European practice guidelines recommend screening individuals with increased metabolic risk using non-invasive methods. Moreover, according to the guidelines, hepatic steatosis should be identified with imaging methods, preferably ultrasound (US), because it is more widely available and cheaper than the gold standard, MRI [2].

Although liver biopsy is considered the most accurate method to diagnose hepatic steatosis, it has multiple drawbacks, including sampling only a small portion of the parenchyma, non-negligible risk of complications, and limited accessibility. Therefore, clinical practice has shifted towards non-invasive imaging techniques to detect fatty liver, as these are more readily available, put less burden on the patient, and can be used to assess focal variations in fat content [3]. Grayscale US is an efficient method to diagnose hepatic steatosis based on well-established morphological signs such as increased liver reflectivity, distal attenuation of US signal, blurring of hepatic vessels and gallbladder wall, or focal sparing at typical locations. The disadvantages of grayscale US are its relatively weak sensitivity for lower grades (<20%) of steatosis, the difficulties with scanning morbidly obese patients with body mass index (BMI) > 40 kg/m^2^, and considerable dependence on the observer’s experience [4]. Computed tomography (CT) is an alternative imaging technique that could quantify hepatic steatosis with good accuracy; however, it exposes patients to a significant amount of ionizing radiation [5]. MRI is the most sensitive imaging method, which can reliably detect even low-grade, between 5% and 10%, steatosis. MRI-PDFF has become a universally accepted reference technique as it can stage steatosis with accuracy comparable to liver biopsy; also, the entire liver can be evaluated with a single scan [6]. However, MRI-PDFF’s high cost and limited availability do not allow screening of large patient populations.

Semi-quantitative scores such as the ultrasonographic fatty liver indicator (US-FLI) can improve the reproducibility of US diagnosis, and identify patients who have non-alcoholic steatohepatitis (NASH) [7]. Another semi-quantitative metric is the hepatorenal index (HRI), which is the ratio between the brightness of the liver and the right renal cortex on grayscale US. HRI is less operator-dependent, and it has been shown to have a good detection rate of mild and even better detection of moderate and severe steatosis [8,9,10]. Meanwhile, the calculation of HRI can be time-consuming, and selecting a region of interest (ROI) can be subjective, weakening the measurement’s reproducibility. Recently, artificial intelligence-calculated HRI (AI-HRI) measurement has become available, where pixels of the liver and renal cortex are delineated by a deep convolutional neural network (DCNN), and positioning of the ROIs, and calculation of the HRI are fully-automated [11]. Furthermore, multiple quantitative ultrasound (QUS) parameters, which allow for simultaneous assessment of liver fibrosis, inflammation, and steatosis in chronic liver diseases, can be measured on advanced systems [12,13,14]. The performance of some of the QUS metrics, such as the ultrasound attenuation coefficient (AC) and ultrasound backscatter-distribution coefficient (SC), has been very good in the classification of all steatosis grades according to multiple studies [15,16].

In the present study, we have used AI-HRI to diagnose hepatic steatosis in NAFLD patients and evaluated its interobserver reproducibility and diagnostic accuracy using MRI-PDFF as the reference method. According to our knowledge, this is the first study that has directly compared AI-HRI with other US parameters for classifying steatosis grades.

## 2. Materials and Methods

### 2.1. Patients

This single-center prospective cohort study was approved by the regional and institutional committee of science and research ethics of our university and written informed consent was obtained from all participants according to the World Medical Association Declaration of Helsinki, revised in Edinburgh in 2000. We prospectively enrolled 271 participants who were examined for suspected liver steatosis in our institution between July 2020 and November 2022. The eligibility criteria to participate in the study included the following: 18 years or older, referral to an imaging study to rule out clinically suspected hepatic steatosis, completed artificial intelligence augmented HRI and MRI-PDFF measurements of liver fat content, clinical findings consistent with NAFLD based on the diagnostic criteria of the European Clinical Practice guidelines [2]. The participants’ demographic data, including the history of alcohol consumption, were collected from a personal survey, and the medical history and laboratory tests were collected from electronic medical reports. We excluded participants who reported daily alcohol consumption in excess of 20 g (2 drinks) for females or 30 g (3 drinks) for males in the last two years, patients with a history of chronic liver disease due to an etiology other than NAFLD, including chronic viral hepatitis, autoimmune hepatitis, PBC, PSC, long term use of hepatotoxic drugs, as well as patients whose liver iron content was above the normal range (≥2 mg/g) or had a positive genetic test for hereditary hemochromatosis. Patients with acute liver failure (ALF), acute on chronic liver failure (ACLF), decompensated liver cirrhosis (DLC: hepatic encephalopathy, moderate ascites, esophageal bleeding), or extrahepatic biliary obstruction were excluded (Figure 1).

The final patient cohort included 102 subjects (50 females and 52 males) who fulfilled the eligibility criteria, did not meet any exclusion criteria, and had 108 valid ultrasonography and MRI measurements of hepatic steatosis. There were six patients who were followed for chronic liver disease and had US and MRI scans during the study period at two different time points with a 6-month interval. The participants’ mean age (±standard deviation, SD) was 55 ± 13 years. Among the participants, 23 (23/102, 23%) had type 2 diabetes mellitus (T2DM), and 31 (31/102, 30%) were severely overweight with a BMI ≥ 30 kg/m^2^. Demographic information and results of laboratory tests in the patient cohort are summarized in Table 1.

### 2.2. Ultrasound Scanning and AI-HRI Measurements

Patients were asked to fast for at least four hours before the ultrasound scan. We used a Samsung RS85 Prestige ultrasound system (Samsung Medison Co. Ltd., Hongcheon, Republic of Korea) equipped with a CA 1-7S convex probe to scan all participants. The ultrasound scans were performed by an expert radiologist with more than ten years of experience in abdominal ultrasound. The AI-HRI model used in this study has been trained and validated by Cha et al. on pre-transplantation liver US scans as has been reported previously [11]. For AI-HRI measurements, a right intercostal or subcostal view was obtained in supine patients and showing the longitudinal cross-section of the right kidney and the right liver lobe. Then, the EzHRI^TM^ application was selected, which automatically detected the outlines of the renal cortex and the liver parenchyma based on a DCNN image segmentation, and placed two identical-size circular ROIs in the liver and the renal cortex at the same depth from the skin surface avoiding large vessels and high-intensity areas of the medulla. The HRI was calculated from the ratio of average pixel intensity values between the two ROIs (Figure 1). The average time to complete a single measurement was less than ten seconds. The AI-HRI value in each case was the median of repeated measurements on five different images. In the few instances, <5% of the measurements, when the algorithm incorrectly placed an ROI, i.e., in an area concealed by rib shadows, the ROI was repositioned manually. The image acquisition parameters, including gain, dynamic range, and focal depth were selected by the examiner to obtain the best available image quality. If the average pixel intensity value was below ten units in any ROI, the gain was increased to avoid large variations in the intensity ratios. A second examiner, a radiology trainee with more than four years of experience in abdominal ultrasound also measured the AI-HRI in 68 patients on the same day. Both examiners were blinded from each other’s and the patients’ prior results.

### 2.3. Measurement of AC, SC, and Semi-Quantitative Scoring of Hepatic Steatosis on US Images

Together with AI-HRI, during the same scan, we also measured two additional QUS parameters, the AC and the SC, as alternative biomarkers of hepatic steatosis. The detailed protocols of the AC and SC measurements have been published previously [14]. Briefly, the right lobe of the liver was visualized from an intercostal window, and from the QUS application of the scanner, either tissue attenuation imaging (TAI^TM^) or tissue scatter distribution imaging (TSI^TM^) mode was selected. Then, the examiner placed a color-coded ROI in the liver parenchyma and recorded the mean AC in dB/cm/MHz units or the mean SC in arbitrary units. AC measurements with an R-squared (R^2^) value > 0.6 were considered unreliable and discarded. Finally, we calculated the median of valid AC and SC measurements and used them in consecutive analyses.

We also took images of the liver in standard views and used them to calculate the ultrasonographic fatty liver indicator (US-FLI) score [7]. The US-FLI is a semi-quantitative scoring system for grading hepatic steatosis based on B-mode images. The US-FLI is a sum of scores given on multiple different features. A trainee and expert radiologist performed the scoring individually, and the final score reflected the consensus between the examiners. The most important feature was the liver/kidney contrast, which could be absent (score 0), mild/moderate (score 1), or severe (score 2). Other features included the posterior attenuation of the beam, vessel blurring, difficult visualization of the gallbladder wall, difficult visualization of the diaphragm, and areas of focal sparing; each of these was scored as either absent (score 0) or present (score 1). We calculated the US-FLI score for 107 US examinations of 103 participants based on archived images retrieved from our PACS system. The US-FLI score of the patients ranged from 0 to 8; a score ≥ 2 was needed for the diagnosis of steatosis, while a score ≥ 4 was indicative of non-alcoholic steatohepatitis (NASH).

### 2.4. Measurement of the MRI-PDFF

The MRQuantif examination protocol and software (https://imagemed.univ-rennes1.fr/en/mrquantif (accessed on 16 September 2022) were used to measure the MRI-PDFF in the participants’ livers [18]. During the MRI scan, two-dimensional (2D) axial images of the liver at the level of the porta hepatis were acquired with a multi-echo gradient echo sequence, which included twelve echoes with gradually spaced echo times (TE) starting from 1.2 msec with 1.2 msec increments. Other scanning parameters included a repetition time (TR) of 120 msec, a flip angle (FA) of 20 degrees, a pixel bandwidth (Bw) of 2712 Hz, a field-of-view of 400 × 350 mm, a reconstruction matrix of 128 × 116 pixels, and an interslice gap of 10 mm. Each slab was scanned during a single breath-hold of 18 s or less. All participants were scanned with the same Philips Ingenia^TM^ 1.5 T MRI scanner (Philips Healthcare, Amsterdam, the Netherlands) using the Q-Body coil. The software calculated the R2* and the MRI-PDFF using a magnitude-based exponential decay model integrating the variation of the signal linked to the six main fat peaks determined by Hamilton et al. [19]. For visual reference, we performed a complex-based estimation of MRI-PDFF and reconstructed color-coded fat fraction maps in a selection of cases using Matlab (The Mathworks, Natick, MA, USA) code (https://github.com/welcheb/FattyRiot (accessed on 30 September 2020) of Berglund et al. [20] (Figure 2). The MRI scans were completed within a month after the ultrasound scans and evaluated blinded from US results. Similar to previous publications, we classified patients into four severity grades (none: <5%, mild: 5–10%, moderate: 10–20%, severe: ≥20%) based on the amount of hepatic steatosis measured with MRI-PDFF [4,14,16,21].

### 2.5. Statistical Analysis

We used the Shapiro–Wilk test to confirm normal distribution of continuous demographic, biochemical, and imaging variables. The analysis of variance (ANOVA) and post-hoc Tukey’s honestly significant difference (HSD) tests were applied to compare continuous variables with normal distributions (i.e., AC and SC) between multiple groups. We used the Kruskal–Wallis rank sum test and, post hoc, the pairwise Wilcoxon rank sum test to compare not normally distributed variables (i.e., AI-HRI and US-FLI). We adjusted *p*-values with the Benjamini–Hochberg method to control the false discovery rate (FDR) in multiple comparisons. The Spearman’s rank correlation coefficient (r_s)_ was calculated to assess the strength of the relationship between different image-based biomarkers of hepatic steatosis and between the AI-HRI measurements of the two examiners. We constructed a Bland–Altman plot to evaluate the interobserver agreement between the AI-HRI values measured by two examiners and calculated the intraclass correlation coefficient (ICC) using a two-way mixed effect model.

We built multiple univariable regression models to identify significant associations between clinical variables and AI-HRI. The age, gender, weight, height, BMI, liver-to-skin distance, type 2 diabetes, blood glucose, hematocrit, platelet count, international normalized ratio (INR), serum albumin, aspartate aminotransferase (AST), alanine aminotransferase (ALT), alkaline phosphatase (ALP), γ-glutamil transferase (GGT), total bilirubin, blood urea nitrogen (BUN), serum creatinine were tested as independent variables against AI-HRI as the dependent variable. Clinical factors achieving significance in the univariable analysis were also evaluated in a multivariable model. The adjusted R-squared (R^2^) metric was used to assess the strength of the associations.

We performed receiver operating characteristic (ROC) curve analyses to assess the performance of the different diagnostic methods of hepatic steatosis with MRI-PDFF used as the reference method. We also calculated multiple performance metrics, including area under the ROC curve (AUC), sensitivity (sens.), specificity (spec.), negative predictive value (NPV), positive predictive value (PPV), and accuracy (acc.). The thresholds of AI-HRI, which could accurately differentiate between consecutive steatosis grades, were also calculated. The Delong test was used to compare the AUC values between different hepatic steatosis metrics. A ROC curve power analysis was also completed using the formula described by Obuchowski et al. to estimate the smallest sample size that allows for accurate discrimination between categories with a type I error rate < 0.05 and a type II error rate < 0.2 [22].

Continuous variables were reported in a mean and standard deviation (SD) format, and categorical variables as numbers and percentages. The r_s_, ICC, and AUC values were reported as median and 95% confidence intervals (CI). We applied a *p* < 0.05 threshold to declare statistical significance in all comparisons. We performed all statistical analysis with the RStudio 2022.07.2 software package (https://rstudio.com (accessed on 2 December 2022).

## 3. Results

### 3.1. Comparison of AI-HRI and Other QUS Parameters between Different Grades of Steatosis

We measured multiple QUS parameters, AI-HRI, AC, and SC, and calculated the semi-quantitative US-FLI score to determine the severity of steatosis in 108 independent measurements in 102 NAFLD patients. The study cohort consisted of 30 cases (30/108, 27.7%) without steatosis (<5% MRI-PDFF), 24 cases (24/108, 22.2%) of mild (from 5% ≤ to <10% MRI-PDFF), 37 cases (37/108, 34.3%) of moderate (from 10% ≤ to <20% MRI-PDFF) and 17 cases (17/108, 15.7%) of severe (≥20% MRI-PDFF) steatosis. The AI-HRI ranged from 0.45 to 5.90, with a mean of 2.27 ± 0.96 for all participants. The AI-HRI was significantly higher in mild steatosis (2.155 ± 0.776) compared to the normal liver (1.480 ± 0.607, *p* < 0.003) and was elevated in moderate steatosis (2.777 ± 0.923, *p* < 0.018) compared to mild steatosis. However, the AI-HRI values were not significantly different between moderate and severe steatosis (2.711 ± 0.822, *p* < 0.787) (Figure 3). The US-FLI score was not significantly higher in mild steatosis (0.900 ± 1.398) compared to the normal liver (1.542 ± 1.318, *p* < 0.074), but it showed significant increase both in moderate (4.000 ± 1.509, *p* < 0.001) and in severe (5.941 ± 0.966, *p* < 0.001) steatosis compared to lower grades. The AC values showed significant gradual increase through all consecutive steatosis grades. Meanwhile, SC was significantly different only between normal liver (91.75 ± 11.03) and mild (101.37 ± 7.15, *p* < 0.001) steatosis (Table 2). No adverse events occurred during the patient scans.

### 3.2. Evaluation of AI-HRI for Detection of Different Grades of Hepatic Steatosis

We performed a ROC curve analysis with AI-HRI, AC, SC, and US-FLI to evaluate the performance of these methods in differentiating normal liver (<5% MRI-PDFF) from mild (≥5% MRI-PDFF) steatosis. The MRI-PDFF was used as the reference method to determine steatosis grades. AI-HRI was able to classify patients into normal and mild steatosis groups with fair accuracy (AUC = 0.758, 95% CI = 0.621–0.894) (Figure 4). We also calculated the spec., sens., PPV, NPV, and acc. metrics for thresholds enabled the most accurate classification (Table 3). The AC (AUC = 0.829, 95% CI = 0.713–0.945, *p* < 0.281) performed relatively better, while SC (AUC = 0.772, 95% CI = 0.645–0.898, *p* < 0.851) similar, and US-FLI (AUC = 0.639, 95% CI = 0.497–0.781, *p* < 0.175) relatively worse than AI-HRI in the same classification task. Due to the low correct prediction rate, the ROC analysis performed with US-FLI had limited statistical power (true positives = 43.6%). The probability of type II error was <20% in all other ROC analyses with the current sample size.

We also evaluated the same US methods for the classification of absent and mild (<10% MRI-PDFF) versus moderate and severe (≥10% MRI-PDFF) steatosis. The AI-HRI could differentiate between advanced and mild or absent steatosis (AUC = 0.803, 0.721–0.885) with good accuracy. The SC (AUC = 0.805, 95% CI = 0.720–0.890, *p* < 0.997), performed very similarly to AI-HRI in the second classification. Meanwhile, both AC (AUC = 0.895, 0.835–0.955, *p* < 0.031) and US-FLI (AUC = 0.937, 95% CI = 0.898–0.975, *p* < 0.002) significantly outperformed AI-HRI in diagnostic accuracy for advanced steatosis. The performance of AI-HRI was not significantly different (*p* < 0.578) between the classification problems. The power for the detection of moderate/severe steatosis was above 90% in the case of all tested diagnostic methods.

### 3.3. Correlation of AI-HRI with Other Methods

We found moderate but significant positive correlation between AI-HRI and MRI-PDFF (r_s_ = 0.528, 95% CI = 0.377–0.651, *p* < 0.001), as well as between AI-HRI and US-FLI measurements (r_s_ = 0.498, 95% CI = 0.329–0.635, *p* < 0.001) (Figure 5). The US-FLI values showed very strong significant correlation with MRI-PDFF (r_s_ = 0.804, 95% CI = 0.706–0.863, *p* < 0.001) and a strong correlation with AC (r_s_ = 0.690, 95% CI = 0.565–0.782, *p* < 0.001). The correlation was also strong between AC and MRI-PDFF (r_s_ = 0.775, 95% CI = 0.660–0.849, *p* < 0.001), but only moderate, although significant, between AC and AI-HRI (r_s_ = 0.597, 95% CI = 0.464–0.700, *p* < 0.001). SC showed strong correlation with MRI-PDFF (r_s_ = 0.6, 95% CI = 0.442–0.724, *p* < 0.001) but only moderate with AI-HRI (r_s_ = 0.473, 95% CI = 0.296–0.621, *p* < 0.001).

In the univariable regression analysis, four independent variables, including age (R^2^ = 0.044, *p* < 0.0395), BUN (R^2^ = 0.103, *p* < 0.004), height (R^2^ = 0.042, *p* < 0.0368), and INR (R^2^ = 0.057, *p* < 0.0333) showed very weak but significant association with AI-HRI. In the multivariable model, none of these factors was a significant predictor of AI-HRI.

### 3.4. Interobserver Agreement of AI-HRI Measurements

There was a moderate but significant correlation between AI-HRI measurements performed by the two examiners (r_s_ = 0.572, 95% CI = 0.340–0.728, *p* < 0.001). The ICC was 0.635 (95% CI = 0.411–0.774, *p* < 0.001), which indicated good interobserver agreement. The Bland–Altman analysis revealed −10.29% average bias between the examiners, while the limits of agreement (LOA) were at 57.70% and −78.28% (Figure 5).

## 4. Discussion

In the present study, we have demonstrated that AI-HRI is a reliable method for the diagnosis and classification of hepatic steatosis in NAFLD. In previous studies, which evaluated HRI, the liver and kidney ROIs were manually selected, which caused significant differences between the diagnostic protocols [8,9,10,23,24].

In previous studies evaluating non-invasive biomarkers of steatosis, 5%, 10%, and 20% MRI-PDFF were used as optimal thresholds for diagnosing mild, moderate, and severe hepatic steatosis, respectively, as these cutoff values closely approximate the classification into S1, S2, and S3 histology grades [6,16,21,25]. We think that for the unambiguous comparability of our results with other non-invasive diagnostic techniques, it was important to use the same classification for steatosis severity as in previous studies.

The accuracy of manually labeled HRI in the detection of mild steatosis showed considerable variability between different studies, with AUC values ranging between 0.68 and 0.92 [8,9,10]. The performance of AI-HRI was within the above range; however, its AUC of 0.76 was superior to some recently published results obtained with high-end ultrasound systems, which measured HRI manually [8]. The best diagnostic threshold for the detection of mild steatosis was 1.53, which is almost identical to the threshold at 1.54 reported for manual HRI in a study that compared US metrics with magnetic resonance spectroscopy proton density fat fraction (MRS-PDFF). Using nearly identical thresholds, the sensitivity was higher (83% vs. 50%), and the specificity was lower (67% vs. 92%) with AI-HRI than with HRI. In another study comparing HRI with histology grades, the cutoff value for mild steatosis was 1.46, which had only 43% sensitivity and 91% specificity. These data suggest that AI-HRI may outperform conventional HRI in detecting low-grade hepatic steatosis as it has greater sensitivity at similar thresholds. Thus, AI-HRI could be an efficient tool for screening patients with suspected NAFLD.

The AI-HRI performed slightly better in diagnosing at least moderate steatosis, although its AUC of 0.81 was not significantly different from the AUC calculated for mild steatosis. The accuracy of AI-HRI was also better compared to manually labeled HRI, which had an AUC of 0.71 and 0.74 for the detections of moderate and severe S2 and S3 histology grade steatosis, respectively [8]. The sensitivity of AI-HRI was again better (70% vs. 47–52%), and its specificity is lower (80% vs. 85–94%) compared to HRI. Meanwhile, the diagnostic threshold for AI-HRI was considerably higher (2.25 vs. 1.48 and 1.79). However, the direct comparison between the two studies is difficult as they used different reference methods, and the exact relationship between MRI-PDFF and steatosis grade detected with histology is still undetermined [6].

The range of AI-HRI (0.45–5.90) was comparable to the conventional HRI (0.77–4.2) as reported previously [9]. The mean AI-HRI (2.27) in the study cohort was higher than the mean HRI (1.4–1.56) in previous studies. This can be mainly attributed to the lower percentage of patients without significant steatosis (27.7%) in our study cohort compared to patient populations in previous studies where the percentage of negative cases (47.8–69.2%) was much higher [8,9,10]. We also found significant differences between AI-HRI measured in normal liver, mild steatosis, and mild and moderate steatosis, indicating robust diagnostic performance.

The agreement between AI-HRI measured by two examiners (ICC = 0.64) was good, and it was in the range of the interobserver agreement reported for manually labeled HRI (ICC = 0.58–0.68) [11]. The correlation between the two observers’ measurements was weaker with AI-HRI than with conventional HRI (r_s_ =0.57 vs. Pearson’s r =0.70). Meanwhile, the mean interobserver bias of −10% and the LOA of −78% and 58% in a Bland–Altman analysis were all higher compared to the bias of 2% and LOA of −47% and 51% reported for HRI [10]. The disagreement between the results can be partly explained by the differences in the study protocol. In our investigation, the mean of five repeated measurements was recorded by both examiners as the AI-HRI. In contrast, in the other study, HRI measurements were preselected, and only the three closest values with a difference of less than 0.2 were used to calculate the mean HRI, which could reduce interobserver variability. In addition, our study cohort included relatively higher numbers of obese patients, the mean BMI was 29 kg/m^2^ vs. 23 kg/m^2^, which could also influence the reproducibility.

Our study is the first to directly compare AI-HRI with other QUS methods for diagnosing hepatic steatosis. The AC was the most accurate in diagnosing all steatosis grades, with excellent and good prediction rates for moderate (AUC = 0.90) and mild steatosis (AUC = 0.83), respectively. A potential drawback of AC is the relatively small difference between diagnostic thresholds for mild (AC = 0.77) and moderate steatosis (AC = 0.83), which can cause miss diagnosis if the interobserver variation is large, especially in difficult-to-scan patients. The SC’s performance was very similar to AI-HRI in the classification of steatosis. Meanwhile, US-FLI, which relies on semi-quantitative scoring of US signs on grayscale images, had a low detection rate for mild steatosis (AUC = 0.64, and sens. = 58%) but high for moderate to severe steatosis (AUC = 0.94, and sens. = 67%). These findings are in line with reports, which indicated that the sensitivity of grayscale US is poor for detecting mild steatosis and excellent for high-grade steatosis [26,27]. We also agree with Petzold et al. that AI-HRI should be evaluated together with grayscale US signs, as these can identify patients with high-grade steatosis with greater accuracy [8]. However, our study has also clearly demonstrated that AI-HRI is better for detecting low-grade steatosis than grayscale US.

This study has several limitations. First, we did not use histology grading as a reference for steatosis as NAFLD patients, especially those with low-grade steatosis, are seldom biopsied; and recent studies have shown that MRI-PDFF can classify all grades of steatosis with extremely high accuracy and reliability [6,28]. There has been a large amount of evidence published in multiple papers that MRI-PDFF is a quantitative noninvasive biomarker that objectively estimates liver fat content providing values over the entire range of biologically relevant liver fat content and, thus, it can be used as a surrogate marker for liver biopsy in clinical studies [3]. Moreover, liver biopsy is not routinely recommended by current European guidelines for diagnosing NAFLD [2]. Therefore, we think that using MRI-PDFF as a reference standard of AI-HRI in our study is rational. Second, a single ultrasound scanner was used for all patient examinations, and the manufacturer trained the DCNN algorithm, which generated the AI-HRI measurements. Other AI software may have very different diagnostic capabilities. Meanwhile, Cha et al. have already evaluated the precision of the same DCNN algorithm and concluded that it achieved similar performance to radiologists for calculating HRI in normal livers and mild steatosis [11]. Third, this is a single-center prospective study conducted in a relatively small patient cohort of 102 subjects. Thus, our result cannot be generalized, and further studies in larger patient groups and preferably in a multi-center setting are required to demonstrate the advantages of AI-HRI in routine clinical practice. Meanwhile, the sample size of 108 independent paired US and MRI-PDFF measurements analyzed in our study is well comparable to other single-center studies investigating the diagnostic performance of the non-invasive diagnostic techniques of steatosis, such as the reports by Caussy et al. on controlled attenuation parameter (CAP), and Jeon et al. on AC, and SC conducted in cohorts of 119 and 120 NAFLD patients, respectively [16,21].

## 5. Conclusions

New, AI-based image analysis techniques can transform US diagnostics by automating the collection of quantitative biomarkers. The AI-HRI is an algorithm developed for automated measurement of HRI on grayscale US. The most significant advantages of AI-HRI are the much shorter measurement time, the reduced workload of the examiners as there is no need for external image processing, the straightforward interpretation of the results, and the uniformity of the diagnostic protocol across all institutions using the same software. The results of our investigation have shown that AI-HRI could detect mild and moderate steatosis with good diagnostic accuracy. AI-HRI has shown great potential as a fast and objective screening tool for detecting hepatic steatosis as it had 83.3% NPV at the 1.53 suggested cutoff value, much higher compared to the 68.8% NPV of the grayscale US signs. The reproducibility of AI-HRI was similar to conventional HRI measurement. Therefore, AI-HRI may be efficiently used to screen large populations for NAFLD and follow up on disease severity.

## Figures and Tables

**Figure 1 medicina-59-00469-f001:**
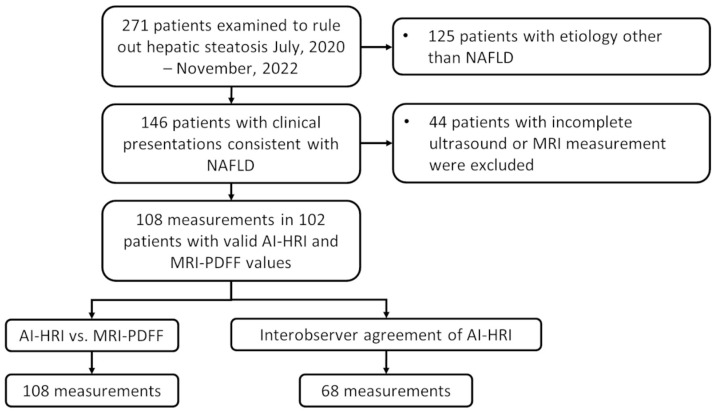
The flowchart demonstrates patient selection and study design. We prospectively enrolled 102 participants with suspected non-alcoholic fatty liver disease (NAFLD) in our study. Two hundred and seventy-one patients were referred to either an ultrasound or an MRI scan at our department to rule out hepatic steatosis; out of these, in 146 patients who did not show signs of acute liver failure, clinical findings were consistent with NAFLD. Forty-four patients who did not have a complete artificial intelligence-calculated hepatorenal index (AI-HRI) and magnetic resonance imaging proton density fat fraction (MRI-PDFF) measurements were excluded from the study. We evaluated the diagnostic accuracy of AI-HRI by comparing it to MRI-PDFF as a reference using 108 independent measurements of 102 patients. The interobserver agreement of AI-HRI was assessed in 68 cases, where measurements by two different examiners were available.

**Figure 2 medicina-59-00469-f002:**
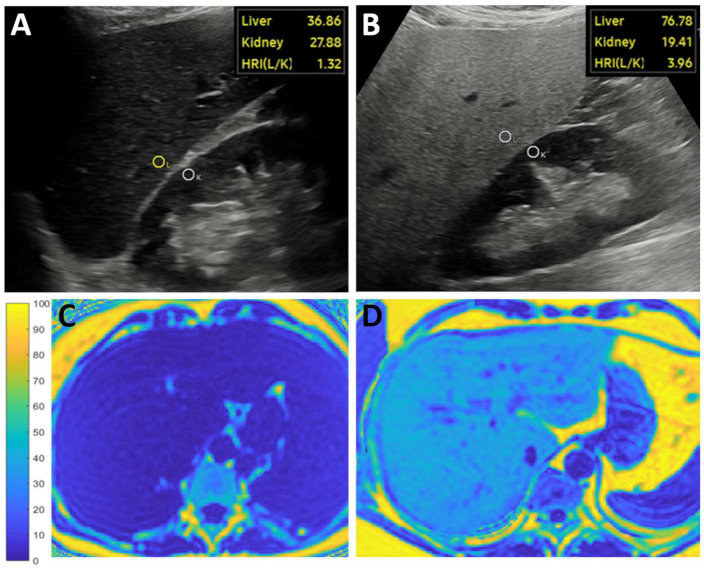
Representative images show measurements of the artificial intelligence-calculated hepatorenal index (HRI) and magnetic resonance imaging fat fraction (MRI-PDFF). The examiner took a longitudinal brightness mode image of the right liver lobe and the right kidney. The HRI was calculated from the pixel intensity ratio of two circular regions of interest (ROI) automatically placed in the liver (L) and the kidney cortex (K) by the software after segmentation of the ultrasound image with a deep convolutional neural network. (**A**) The brightness of a non-steatotic liver was similar to the kidney’s cortex resulting in a low HRI. (**B**) The brightness of a severely steatotic liver was much higher than the kidney’s cortex, causing an elevated HRI in a patient diagnosed with NAFLD. The MRI-PDFF maps reconstructed with a complex method were used as a reference. (**C**) In the first cases, the blue color of a non-steatotic liver corresponded to <5% MRI-PDFF on the scale ranging from 0–100%. (**D**) Meanwhile, severe steatosis (≥20% MRI-PDFF) was indicated by the turquoise color of the liver in the second case.

**Figure 3 medicina-59-00469-f003:**
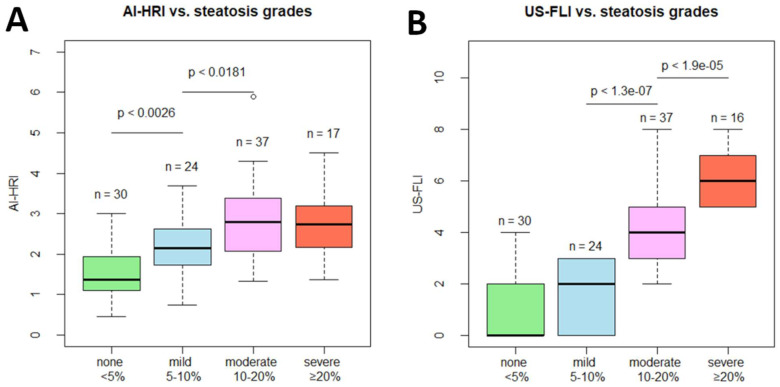
Box plots show the distribution of (**A**) artificial intelligence-calculated hepatorenal index (AI-HRI) values and (**B**) ultrasonographic fatty liver indicator (US-FLI) scores in increasing grades of hepatic steatosis. Magnetic resonance imaging proton density fat fraction (MRI-PDFF) was used as the reference method. We compared AI-HRI and US-FLI between different steatosis grades with the pairwise Wilcoxon rank sum test and labeled significant differences with the *p*-value.

**Figure 4 medicina-59-00469-f004:**
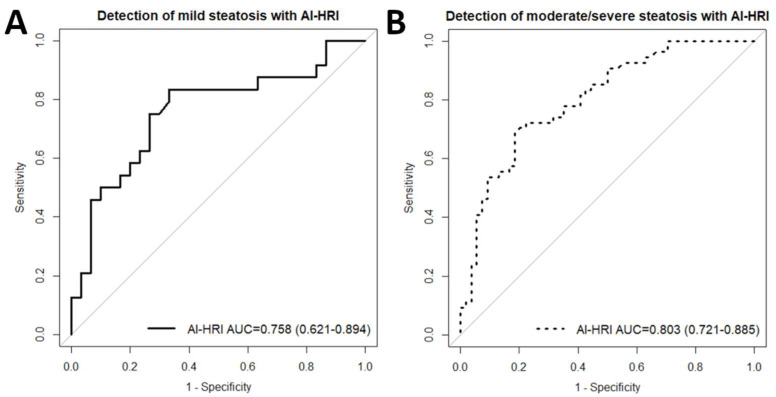
(**A**) The plot shows receiver operating characteristics (ROC) curve analyses with artificial intelligence-calculated hepatorenal index (AI-HRI) for the classification of normal liver and mild steatosis. The accuracy of AI-HRI was fair based on the area under the curve (AUC) value of 0.758. The 95% confidence intervals of the AUC are listed inside the brackets. (**B**) The plot shows that AI-HRI could detect moderate/severe steatosis with an AUC of 0.803, indicating good classification accuracy. The magnetic resonance imaging fat fraction (MRI-PDFF) was used as the reference method in both classifications.

**Figure 5 medicina-59-00469-f005:**
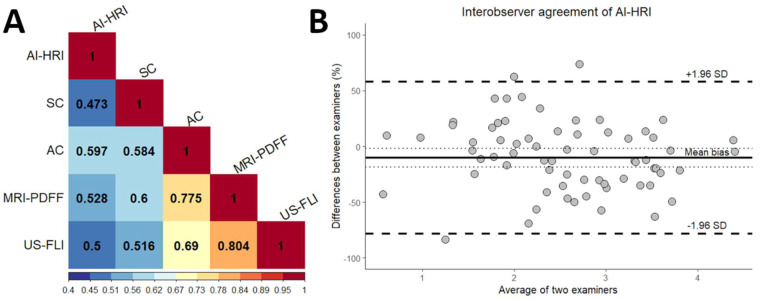
(**A**) The correlation matrix between imaging methods is displayed on a colored heat map. The amount of steatosis was determined with quantitative ultrasound (QUS) metrics, including artificial intelligence-calculated hepatorenal index (AI-HRI) and ultrasound attenuation coefficient (AC), ultrasound backscatter-distribution coefficient (SC), and semi-quantitative ultrasonographic fatty liver indicator (US-FLI); while magnetic resonance imaging fat fraction (MRI-PDFF) was the reference method. The Spearman’s rank correlation coefficients were calculated from pairwise comparisons. (**B**) The Bland–Altman plot shows interobserver agreement between AI-HRI measured by two examiners. The mean of the bias is labeled with a solid, and the upper and lower limits of agreement are with dashed lines.

**Table 1 medicina-59-00469-t001:** Demographic variables and laboratory tests in the NAFLD patient cohort.

Patient Number:	102
Females/males:	50/52
* Age (years):	55 ± 13
* BMI:	28.95 ± 4.63
T2DM:	23/102 (22.5%)
* Platelet(×10^9^/L):	245.17 ± 67.35
* Albumin (g/L):	43.82 ± 3.66
* AST (IU/L):	37.38 ± 26.57
* ALT (IU/L):	48.80 ± 39.84
* ALP (IU/L):	86.93 ± 48.62
* Total bilirubin (µmol/L):	13.89 ± 7.60
* Sodium (mmol/L):	139.89 ± 2.24
* Creatinine (µmol/L):	78.10 ± 21.35
*^$^ APRI:	0.41 ± 0.28
*^$^ Fibrosis-4 Index:	1.38 ± 0.85
*^$^ NAFLD Fibrosis Score:	1.57 ± 1.65
** HSI:	37.83 ± 6.21

* Values are reported as mean ± standard deviation; ^$^ Clinical and laboratory test indices were calculated using the MDCalc website (www.mdcalc.org (accessed on 2 December 2022)); ** hepatic steatosis index = 8 × (ALT/AST ratio) + BMI (+2, if female; +2, if diabetes mellitus) [17]; BMI: body mass index, T2DM: type 2 diabetes mellitus, AST: aspartate aminotransferase, ALT: alanine aminotransferase, ALP: alkaline phosphatase, APRI: AST to platelet ratio index, HSI: hepatic steatosis index.

**Table 2 medicina-59-00469-t002:** Values measured with QUS methods in increasing grades of hepatic steatosis.

* Steatosis Grade:	None	Mild	Moderate	Severe
AI-HRI	1.480 ± 0.607	2.155 ± 0.776	2.777 ± 0.923	2.711 ± 0.822
** *p*-value<	NA	0.003	0.018	0.787
*** AC (dB/cm/Mhz)	0.674 ± 0.084	0.797 ± 0.089	0.895 ± 0.097	1.004 ± 0.139
** *p*-value<	NA	0.001	0.002	0.003
*** SC	91.75 ± 11.03	101.37 ± 7.15	105.98 ± 5.40	106.17 ± 4.81
** *p*-value<	NA	0.001	0.115	1.00
*** US-FLI	0.900 ± 1.398	1.542 ± 1.318	4.000 ± 1.509	5.941 ± 0.966
** *p*-value<	NA	0.074	0.001	0.001

***** Classification is based on MRI-PDFF as the reference method (none: <5%, mild: 5–10%, moderate: 10–20%, severe: ≥20%), ** Compared to lower grade(s) of steatosis, *** Mean ± standard deviation, AI-HRI: artificial intelligence-calculated hepatorenal index, AC: ultrasound attenuation coefficient, MRI-PDFF: magnetic resonance imaging proton density fat fraction, SC: ultrasound backscatter-distribution coefficient, US-FLI: ultrasonographic fatty liver indicator.

**Table 3 medicina-59-00469-t003:** Performance metrics from ROC analyses with QUS parameters.

Method	Thresh.	Spec.	Sens.	PPV	NPV	Acc.
* For differentiation between normal liver (<5%) and mild (≥5%) steatosis						
AI-HRI (AUC = 0.85)	1.23	0.367	0.875	0.525	0.786	0.593
	** 1.53	0.667	0.833	0.667	0.833	0.741
	1.85	0.733	0.750	0.692	0.786	0.741
AC (AUC = 0.922)	0.74	0.828	0.750	0.783	0.800	0.792
	** 0.77	0.897	0.708	0.850	0.788	0.811
	0.79	0.931	0.583	0.875	0.730	0.774
SC (AUC = 0.860)	90.47	0.414	0.958	0.575	0.923	0.660
	** 93.93	0.552	0.875	0.618	0.842	0.698
	94.87	0.586	0.792	0.613	0.773	0.679
US-FLI (AUC = 0.85)	*** 2	0.733	0.583	0.636	0.688	0.667
* For differentiation between absent/mild (<10%) and moderate/severe (≥10%) steatosis						
AI-HRI (AUC = 0.803)	2.21	0.778	0.722	0.765	0.737	0.750
	** 2.25	0.796	0.704	0.776	0.729	0.750
	2.29	0.815	0.685	0.787	0.721	0.750
AC (AUC = 0.895)	0.81	0.792	0.840	0.792	0.840	0.816
	** 0.83	0.849	0.800	0.833	0.818	0.825
	0.85	0.868	0.760	0.844	0.793	0.816
SC (AUC = 0.805)	98.37	0.547	0.940	0.662	0.906	0.738
	** 100.22	0.623	0.920	0.697	0.892	0.767
	101.35	0.642	0.860	0.694	0.829	0.748
US-FLI (AUC = 0.937)	*** 4	0.944	0.667	0.923	0.739	0.806

* The magnetic resonance imaging fat fraction (MRI-PDFF) was used as the reference method. ** Labels the best thresholds with the highest diagnostic accuracy. *** Diagnostic thresholds of US-FLI for non-alcoholic fatty liver disease (NAFLD) and non-alcoholic steatohepatitis (NASH) were defined by Ballestri et al. [7]. Acc: accuracy, AI-HRI: artificial intelligence-calculated hepatorenal index, AC: ultrasound attenuation coefficient, AUC: area under the ROC curve, NPV: negative predictive value, PPV: positive predictive value, SC: ultrasound backscatter-distribution coefficient, Sens.: sensitivity, Spec.: specificity, Thresh.: threshold, US-FLI: ultrasonographic fatty liver indicator.

## Data Availability

Not applicable.

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
