# Peer review of "Evaluation of Artificial Intelligence-Calculated Hepatorenal Index for Diagnosing Mild and Moderate Hepatic Steatosis in Non-Alcoholic Fatty Liver Disease"

_medicina, 2023, doi:10.3390/medicina59030469_

Round 1
Reviewer 1 Report
The observational study by Szombor et al. aimed to evaluate the hepatorenal index determined using a novel artificial intelligence as a diagnostic tool for hepatic steatosis. Recently, artificial intelligence algorithms have been of great interest in improving diagnostic tests. The following are some proposed suggestions for the manuscript:
· The study cohort comprises 108 measurements performed on 102 subjects. How were the 108 measurements obtained and selected from the 102 participants?
· I suggest presenting first Figure 1 and then Table 1.
· Although the sample size was a consecutive sample in a specified time, please justify with a sample size formula for diagnostic tests a formal calculation.
· Artificial intelligence studies provide more confident results when a large sample size, including the whole spectrum of the disease, is evaluated. Therefore, the main limitation of the present study is precisely the small sample size, as briefly stated in the discussion section. Please consider increasing the sample size or justify using an appropriate statistic formula for sample size calculation.
- How did the manufacturer train the DCNN algorithm? Include a brief description in the methods section.
· The results section is complete; however, consider starting with the answer to the research question (as in section 3.4 of the results) to be more comprehensible.
· Conclusion section: Add and put in context the diagnostic accuracy measures reported to highlight the significance in clinical practice.
· Include a checklist for reporting studies for diagnostic accuracy, such as STARD.
Author Response
Reviewer #1:
The observational study by Zsombor et al. aimed to evaluate the hepatorenal index determined using a novel artificial intelligence as a diagnostic tool for hepatic steatosis. Recently, artificial intelligence algorithms have been of great interest in improving diagnostic tests. The following are some proposed suggestions for the manuscript:
The study cohort comprises 108 measurements performed on 102 subjects. How were the 108 measurements obtained and selected from the 102 participants?
The reviewer has correctly pointed out that our study includes 108 independent measurements collected from 102 patients. The reason for it is that there were six patients who were followed for chronic liver disease and had more than one ultrasound and MRI scan during the study period at different time points with a 6-month interval. We have included this information in the Material and Methods section of the revised manuscript. Please, see page 4, lines 135-137.
I suggest presenting first Figure 1 and then Table 1.
We have changed the order of appearance of Figure 1 and Table 1. in the revised manuscript as the reviewer has suggested. Please, check pages 3 and 4.
Although the sample size was a consecutive sample in a specified time, please justify with a sample size formula for diagnostic tests a formal calculation. Artificial intelligence studies provide more confident results when a large sample size, including the whole spectrum of the disease, is evaluated. Therefore, the main limitation of the present study is precisely the small sample size, as briefly stated in the discussion section. Please consider increasing the sample size or justify using an appropriate statistic formula for sample size calculation.
We consulted the manufacturer’s website (https://www.samsunghealthcare.com/en/LiverAnalysisSolution) on the technical details of the EzHRITM application. The basic concept of the AI-HRI model and the details of the training process have been described in a publication by Cha et al., which we have referenced in the Materials and Methods section (Please, see page 4, lines 154-156). The deep learning model was trained on 294 pre-transplantation liver US scans. Cha et al. have also validated their AI-HRI model and found that its consistency in measuring hepatorenal index was similar to trained radiologists.
It is also important to emphasize that AI-HRI measurements were continuously supervised by the human examiner performing the US scan. In the few instances, <5% of the measurements, when any of the ROIs was incorrectly positioned, i.e. in an area concealed by shadowing by a rib, the human observer could manually correct the position of the AI-selected ROI. Thus, AI-HRI measurements were automated but also fully supervised minimizing diagnostic errors from any training glitches. We added this detail to the Materials and Methods section of the revised manuscript. Please refer to page 5, lines 166-168.
The sample size of 108 independent paired US and MRI-PDFF measurements analyzed in our study is well comparable to other single-center studies investigating the diagnostic performance of the non-invasive diagnostic techniques of steatosis including the reports by Caussy et. al. on controlled attenuation parameter (CAP) and Jeon et. al. on tissue attenuation imaging (TAI) and tissue background scatter imaging (TSI) conducted on cohorts of 119 and 120 NAFLD patients, respectively.We have included this statement in the Discussion. Please, see page 13, lines 473-478.
We have also performed a statistical power calculation with the current sample size using the formula described by Obuchowski et. al. for the ROC curve analysis. We have referenced the formula of Obuchowski et. al. in the Material and Methods (page 7, line 267). The type II error rate was <20% in all ROC analyses performed with AI-HRI. We have added this information to the Results. Please, see page 9, lines 317-319.
The results section is complete; however, consider starting with the answer to the research question (as in section 3.4 of the results) to be more comprehensible.
We agree with the reviewer that the diagnostic performance of AI-HRI should be prioritized in the Results section. We have changed the order of the sections, and the results of the ROC analysis are now reported in section 3.2 of the revised manuscript. Please, check page 8, line 306.
Conclusion section: Add and put in context the diagnostic accuracy measures reported to highlight the significance in clinical practice.
We agree with the reviewer that the advantages of AI-HRI in routine practice need to be emphasized. The greatest advantages of automated AI-HRI calculation over conventional HRI measurements are the significantly shortened examination time, the uniformity of measurements, and that AI-HRI measurement can be completed in one simple step during a routine liver US and it does not require external image processing. AI-HRI has great potential as a fast and objective screening tool for detecting hepatic steatosis as it had 83.3% NPV at the 1.53 suggested cutoff value, much higher than the 68.8% NPV of the grayscale US signs. We have added these data to the Conclusion of the revised manuscript. Please, see page 13, lines 482-490.
Include a checklist for reporting studies for diagnostic accuracy, such as STARD.
We have completed a point-by-point revision of the manuscript using the STARD checklist. Please, see it below:
TITLE OR ABSTRACT
1 Identification as a study of diagnostic accuracy using at least one measure of accuracy
(such as sensitivity, specificity, predictive values, or AUC)
Yes, AUC values with 95% confidence intervals are included (page 1, lines 34-35)
ABSTRACT
2 Structured summary of study design, methods, results, and conclusions
(for specific guidance, see STARD for Abstracts)
Yes, the abstract is structured to Background&Objectives, Materials&Methods, Results and Conclusions sections.Please, check page 1.
INTRODUCTION
3 Scientific and clinical background, including the intended use and clinical role of the index test
Yes, the scientific and clinical background of the study, intended use, and index tests are included. Please, check the Introduction on pages 1 and 2.
4 Study objectives and hypotheses
Yes, study objectives and hypothesis are included (page 2, lines 90-94).
METHODS
Study design 5 Whether data collection was planned before the index test and reference standard
were performed (prospective study) or after (retrospective study)
Yes, data collection was planned before index and reference measurements following a prospective study design (page 3, lines 100-102)
Participants
6 Eligibility criteria
-Yes, eligibility and exclusion criteria are included (page 3, lines 102-117).
7 On what basis potentially eligible participants were identified
(such as symptoms, results from previous tests, inclusion in registry)
- Yes, patients with complete index and reference measurements and clinical background consistent with NAFLD (page 3, lines 104-106).
8 Where and when potentially eligible participants were identified (setting, location and dates)
-Yes, the settings, the time frame, and the location of the study are included (page
3, lines 100-102)
9 Whether participants formed a consecutive, random or convenience series
-Yes, participants formed a consecutive series, and inclusion and exclusion criteria were uniformly applied to all patients (page 3, lines 102-117).
Test methods 10a Index test, in sufficient detail, to allow replication
-Yes, a detailed description of the examination methods is included (page 4, section 2.2).
10b Reference standard, in sufficient detail to allow replication
-Yes, MRI-PDFF was used as the reference standard (page 6, section 2.4).
11 Rationale for choosing the reference standard (if alternatives exist)
-Yes, the choice of MRI-PDFF as reference method is explained in detail (pages 12-13, lines 458-465).
12a Definition of and rationale for test positivity cut-offs or result categories
of the index test, distinguishing pre-specified from exploratory
-Yes, the rationale of classifying cases into mild, moderate, and severe steatosis grades is explained (page 11, lines 387-392).
12b Definition of and rationale for test positivity cut-offs or result categories
of the reference standard, distinguishing pre-specified from exploratory
-Yes, the selection of 5%, 10%, and 20% PDFF cutoff values is explained (page 6, 231-233)
13a Whether clinical information and reference standard results were available
to the performers/readers of the index test
-Yes, the US examiners were blinded from results of the reference measurements (page 6, lines 172-173).
13b Whether clinical information and index test results were available
to the assessors of the reference standard
-Yes, the reference measurements were performed without prior knowledge of clinical status or US results (page 6, lines 229-230).
Analysis 14 Methods for estimating or comparing measures of diagnostic accuracy
15 How indeterminate index test or reference standard results were handled
Yes, multiple index test measurements were performed, and the median value was calculated (page 5, lines 164-165).
16 How missing data on the index test and reference standard were handled
Yes, only patients with a complete US measurement were included (page 3, lines 102-117)..
17 Any analyses of variability in diagnostic accuracy, distinguishing pre-specified from exploratory
Yes, diagnostic performance with multiple thresholds is reported (page 9, Table 3.).
18 Intended sample size and how it was determined
Yes, ROC curve power calculation was performed. Type II error was less than 20% with the current sample size (page 7, lines 264-266).
RESULTS
Participants 19 Flow of participants, using a diagram
20 Baseline demographic and clinical characteristics of participants
Yes, baseline demographic and clinical data are reported (page 4, Table 1.).
21a Distribution of severity of disease in those with the target condition
Yes, the distribution of steatosis severity in the cohort is reported (page 7, section 3.1).
21b Distribution of alternative diagnoses in those without the target condition
Yes, only NAFLD patients were included in the selection criteria. Other etiologies of steatosis were excluded (page 3, lines 102-117).
22 Time interval and any clinical interventions between index test and reference standard.
Yes, the time interval between AI-HRI and MRI-PDFF measurements was less than a month (page 6, line 229).
Test results 23 Cross tabulation of the index test results (or their distribution)
by the results of the reference standard
Yes, the correlation between AI-HRI and MRI-PDFF measurements is reported (page 10, section 3.3).
24 Estimates of diagnostic accuracy and their precision (such as 95% confidence intervals)
Yes, the diagnostic accuracy of AI-HRI is evaluated in a ROC analysis, and AUC values are reported with 95% confidence intervals (page 8, section 3.2).
25 Any adverse events from performing the index test or the reference standard
Yes, no adverse events occurred during the study (page 7, line 291).
DISCUSSION
26 Study limitations, including sources of potential bias, statistical uncertainty, and generalizability
Yes, study limitations are described (pages 11 and 12, lines 455-478).
27 Implications for practice, including the intended use and clinical role of the index test
Yes, implications for clinical practice are described (page 13, lines 486-492).
OTHER INFORMATION
28 Registration number and name of registry
Not applicable.
29 Where the full study protocol can be accessed
Yes, the ethics committee approval number is provided (page 13, lines 505-509)
30 Sources of funding and other support; the role of funders
Yes, founding information is included (page 13, line 503).

Reviewer 2 Report
In this paper, the authors evaluate the artificial intelligence-calculated hepatorenal index (AI-HRI) using a diagnostic method for hepatic steatosis. This is a very interesting paper however; the authors should provide the readers with a useful and practical approach based on their results. In addition, I think that the liver biopsy as a gold standard regarding the measurement of steatosis grade was necessary as it is mentioned in the limitation.
It should be considered that in most references the grade of steatosis is divided into mild (>30%), moderate (30-60%), and severe (<60%), so the reason why this range of steatosis was studied should be explained in discussion.
Author Response
Reviewer #2:
1. In this paper, the authors evaluate the artificial intelligence-calculated hepatorenal index (AI-HRI) using a diagnostic method for hepatic steatosis. This is a very interesting paper; however, the authors should provide the readers with a useful and practical approach based on their results.
We thank the reviewer for his favorable comments on our manuscript. We agree with the reviewer that the topic of automated HRI calculation should raise significant interest in the field of hepatology.
We also agree with the reviewer that the advantages of AI-HRI in routine practice need to be emphasized. The greatest advantages of automated AI-HRI calculation over conventional HRI measurements are the significantly shortened examination time, the uniformity of measurements, and that AI-HRI measurement can be completed in one simple step during a routine liver US and it does not require external image processing. AI-HRI has great potential as a fast and objective screening tool for detecting hepatic steatosis as it had 83.3% NPV at the 1.53 suggested cutoff value, much higher than the 68.8% NPV achieved by grayscale US signs. We have added these data to the Conclusion of the revised manuscript. Please, see page 13, lines 482-490.
2. In addition, I think that the liver biopsy as a gold standard regarding the measurement of steatosis grade was necessary as it is mentioned in the limitation.
We agree with the reviewer that liver biopsy is currently considered the gold standard of fat quantification. However, there is a large amount of evidence published in multiple papers that MRI-PDFF is a quantitative noninvasive biomarker that objectively estimates liver fat content providing values over the entire range of biologically relevant liver fat content, and thus it can be used as a surrogate marker for liver biopsy in clinical studies. In addition, current European guidelines recommend quantitative fat/water selective MRI as a standard diagnostic technique for steatosis. Also, liver biopsy is not routinely recommended by current European guidelines for diagnosing NAFLD. Therefore, we think that using MRI-PDFF as a reference standard of AI-HRI in our study is rational. We have included this explanation in the Discussion of the revised manuscript. Please, find it on pages 12 and 13, lines 458-464.
3. It should be considered that in most references the grade of steatosis is divided into mild (>30%), moderate (30-60%), and severe (<60%), so the reason why this range of steatosis was studied should be explained in discussion.
The reviewer correctly states that during histological evaluation steatosis is divided into four stages (S0-S3) based on the percentage of liver cells showing signs of intracellular lipid accumulation. Meanwhile, in previous studies evaluating non-invasive biomarkers of steatosis 5%, 10%, and 20% MRI-PDFF were used as optimal thresholds for diagnosing mild, moderate, and severe hepatic steatosis, respectively as these cutoff values closely approximate the classification into S1, S2, and S3 histology grades. We think that for the unambiguous comparability of our results with other non-invasive diagnostic techniques, it is important to use the same classification for steatosis severity as in previous studies. We have added this information to the Discussion of the revised manuscript. Please see page 11, lines 387-392.
